Diversity and ecological structure of vibrios in benthic and pelagic habitats along a latitudinal gradient in the Southwest Atlantic Ocean

Chimetto Tonon Luciane A. 1 2 luciane.chimetto@gmail.com
Silva Bruno Sergio de O. 1
Moreira Ana Paula B. 1
Valle Cecilia 1
Alves Nelson Jr 1
Cavalcanti Giselle 1
Garcia Gizele 1
Lopes Rubens M. 3
Francini-Filho Ronaldo B. 4
de Moura Rodrigo L. 1
Thompson Cristiane C. 1
Thompson Fabiano L. 1 fabiano.thompson@biologia.ufrj.br
1 Laboratory of Microbiology, Institute of Biology, Federal University of Rio de Janeiro (UFRJ) , Rio de Janeiro , Brazil
2 Brazilian Biosciences National Laboratory (LNBio), National Research Center Energy and Materials (CNPEM) , Campinas , Brazil
3 Institute of Oceanography, University of São Paulo (IO-USP) , São Paulo , Brazil
4 Department of Environment and Engineering, Federal University of Paraíba (UFPB) , Rio Tinto, PB , Brazil
Coenye Tom
Electronic publication date: 2015 Feb 10
Publication date: 2015
Volume: 3
Electronic Location ID: e741
Received 2014 Nov 14; Accepted 2015 Jan 5
Copyright: © 2015 Chimetto Tonon et al.
Copyright year: 2015
Copyright holder: Chimetto Tonon et al.
License: This is an open access article distributed under the terms of the Creative Commons Attribution License, which permits unrestricted use, distribution, reproduction and adaptation in any medium and for any purpose provided that it is properly attributed. For attribution, the original author(s), title, publication source (PeerJ) and either DOI or URL of the article must be cited.
License URL: https://creativecommons.org/licenses/by/4.0/

Keywords: Habitats, Vibrio, Corals, AdaptML, Rhodoliths, Benthos, Plankton

Funding: Conselho Nacional de Desenvolvimento Científico e Tecnológico (CNPq) Fundação de Amparo à Pesquisa do Estado do Rio de Janeiro (FAPERJ) Coordenação de Aperfeiçoamento de Pessoal de Nivel Superior (CAPES) Funding was provided by Conselho Nacional de Desenvolvimento Científico e Tecnológico (CNPq), Fundação de Amparo à Pesquisa do Estado do Rio de Janeiro (FAPERJ), and Coordenação de Aperfeiçoamento de Pessoal de Nivel Superior (CAPES). The funders had no role in study design, data collection and analysis, decision to publish, or preparation of the manuscript.

==============================
We analyzed the diversity and population structure of the 775 Vibrio isolates from different locations of the southwestern Atlantic Ocean (SAO), including St. Peter and St. Paul Archipelago (SPSPA), Abrolhos Bank (AB) and the St. Sebastian region (SS), between 2005 and 2010. In this study, 195 novel isolates, obtained from seawater and major benthic organisms (rhodoliths and corals), were compared with a collection of 580 isolates previously characterized (available at www.taxvibrio.lncc.br). The isolates were distributed in 8 major habitat spectra according to AdaptML analysis on the basis of pyrH phylogenetic reconstruction and ecological information, such as isolation source (i.e., corals: Madracis decactis, Mussismilia braziliensis, M. hispida, Phyllogorgia dilatata, Scolymia wellsi; zoanthids: Palythoa caribaeorum, P. variabilis and Zoanthus solanderi; fireworm: Hermodice carunculata; rhodolith; water and sediment) and sampling site regions (SPSPA, AB and SS). Ecologically distinct groups were discerned through AdaptML, which finds phylogenetic groups that are significantly different in their spectra of habitat preferences. Some habitat spectra suggested ecological specialization, with habitat spectra 2, 3, and 4 corresponding to specialization on SPSPA, AB, and SS, respectively. This match between habitat and location may reflect a minor exchange of Vibrio populations between geographically isolated benthic systems. Moreover, we found several widespread Vibrio species predominantly from water column, and different populations of a single Vibrio species from H. carunculata in ecologically distinct groups (H-1 and H-8 respectively). On the other hand, AdaptML detected phylogenetic groups that are found in both the benthos and in open water. The ecological grouping observed suggests dispersal and connectivity between the benthic and pelagic systems in AB. This study is a first attempt to characterize the biogeographic distribution of vibrios in both seawater and several benthic hosts in the SAO. The benthopelagic coupling observed here stands out the importance of vibrios in the global ocean health.

Introduction

The distribution of microorganisms in space does not occur evenly, but generally, it is efficient because local environmental conditions selects specific populations to become relatively more abundant. In situations where this may be a good approximation of the primary mechanisms creating population structure, one can invoke a niche theory to understand how organisms interact with other species and their environment (Tilman, 1982). Under this model, competition with co-existing organisms is an important determinant of which species are present in specific environment (Hibbing et al., 2010). Over evolutionary time scales, competition can cause species to adapt alternative lifestyles, e.g., to attach to different hosts or different substrates in the same hosts. Such trade-offs are a necessary result of constraints on cellular machinery, such as cell-surface proteins and enzymatic capacity or genome size, resulting in specialized populations with limited niche overlap (Preheim, 2010).

Microbes associated with benthic holobionts, such as corals, play key roles in the health of their hosts (Rosenberg et al., 2007). Vibrios represent a significant component of the culturable microbiomes of marine hosts and plankton. Their proliferation and population abundance can be induced by multiple factors including increased water temperature related to global climate change, which can cause impacts in ecosystem structure, including Vibrio-associated diseases (Baker-Austin et al., 2013; Vezzulli et al., 2012). However, when the proliferation of vibrios is not favored, they are able to find refuge in suitable reservoirs to survive, making use of their extensive adaptive capabilities. For instance, vibrios can change from free-swimming cells to “swarmer cells” that prosper in more viscous environments such as biofilms (McCarter, 1999), or switch from an active stage to a dormant and viable but not culturable (VBNC) stage, but yet may still be very potent opportunists if favorable conditions recur (McDougald & Kjelleberg, 2006).

Previous studies suggest that the proliferation of vibrios in the plankton results in lethal vibrio infections in the benthos, suggesting mechanisms of benthopelagic coupling. In other words, these events reveal a causal relationship between the water-column and benthic processes, which may influence the health of the global ocean (Vezzulli et al., 2010; Vezzulli et al., 2012). For instance, an increase in seawater temperature (average range 21.0 °C to 24.3 °C) appears to induce the growth of certain vibrios (e.g., V. harveyi related species) and the concomitant mass mortality of the gastropod Haliotis discus hannai in northern Japan (Fukui, Saitoh & Sawabe, 2010). It has also been shown that the ocean warming observed in the last decades has induced a significant increase in the abundance and range of vibrios in a long-term study in the North Sea (Vezzulli et al., 2012). Moreover, increases of organic material used by Vibrio for energy may be an additional factor in determining Vibrio dynamics. Phosphorous, for example, seems to influence the abundance of planktonic vibrios according to a metagenomic study of bacterioplankton diversity in a tropical bay. According to this study, nutrient limitation effects can be observed at community (metagenomic) and population levels (total prokaryote and vibrio counts) (Gregoracci et al., 2012). Vibrios display a broad metabolic range that enables them to use a wide variety of carbon sources (Thompson & Polz, 2006). Nevertheless, it is not clear whether vibrioplanktonic cells are genetically and ecologically coupled to vibrio cells that live in association with holobionts.

We previously reported massive mortality of the major coral reef builder, Mussismilia braziliensis, and we isolated potential causative agents (Francini-Filho et al., 2008; Alves et al., 2010). According to these former studies, the diseases affecting corals were tissue necrosis in Phyllogorgia dillatata, bleaching in M. hispida and white plague and bleaching in M. braziliensis. Most of the isolated vibrios fell into the Harveyi clade (Sawabe, Kita-Tsukamoto & Thompson, 2007; Sawabe et al., 2013) and V. coralliilyticus. The vibrio isolates of these studies encompassed strains originating from both apparently healthy and diseased corals and had high pathogenic potential for different animals. V. alginolyticus 40B, V. communis 1DA3 and V. coralliilyticus 2DA3 caused 25–88% mortality in the model organism Drosophila melanogaster (Alves et al., 2010). However, the possible ecologic structure and genetic connectivity among vibrios from the coastal-oceanic and benthic-pelagic systems of the SAO remains unclear.

Studies performed in subtropical areas indicate that coastal vibrioplanktonic communities are finely structured in discrete phylogenetic clusters, revealing the co-occurrence of several hundreds of closely related populations (Thompson et al., 2005a; Thompson et al., 2005b). Sympatric differentiation may be due to niche partitioning and specialization, with the association of different groups of bacterioplanktonic species with different habitats (zooplankton, particles and water) in the same geographic location. Hunt et al. (2008) showed that some vibrio species appeared to occur only in association with plankton, whereas other species appeared to be exclusively free-living.

Little is known about the vibrioplankton population diversity and structure in the SAO, and whether vibrioplankton diversity is linked to benthic compartments in this region. Oceanic islands, and reef systems, such as Saint Peter and Saint Paul Archipelago (SPSPA) and Abrolhos Bank (AB) are important environments located in the SAO. SPSPA is constituted by five emerged summits of the Mid-Atlantic Ridge approximately 1,000 km off the coast of Natal (Moreira et al., 2013). It is a biodiversity hotspot. AB is an extension of the continental shelf off the south of the Brazilian State Bahia (17–20°S), corresponding to approximately 45,000 km2 (Amado-Filho et al., 2012). This bank comprises the world’s largest rhodolith bed (aggregates of non-geniculate crustose coralline algae nodules) forming large expanses of hard bottom habitat with approximately 21,000 km2, contributing to the SAO as a nursery place, nutrient producer and carbonate storage (Cavalcanti et al., 2013a; Cavalcanti et al., 2013b). Rhodolith beds stand together with kelp beds, seagrass meadows, and crustose coralline algae reefs as one of the world’s four largest macrophyte-dominated benthic communities (Foster, 2001). AB is a particularly nutrient-rich reef system, with higher nutrients concentration levels than reef systems less influenced by river estuaries (Bruce et al., 2012). SPSPA and AB are under the influence of water masses formed by the warm and salty Tropical Water (TW), with temperature values ranging from 22 to 27 °C, and salinity values ranging from 36.5 to 37 (Castro & Miranda, 1998; Rossi-Wongtschowski et al., 2006). The benthic communities occurring in the different islands may also be locally a source of nutrients in the SAO. SS is a sea passage 25 km long, 2–7 km wide and 40 m maximum depth located between the island of São Sebastião (municipality of Ilhabela) and the mainland (São Sebastião), on the coast upstate São Paulo, southeastern Brazil. The currents in the channel are directed by the wind and the water temperature range is 15–28 °C. The region is influenced by both the warm waters of the Brazil Current (22–28 °C) that goes down to the south and the cold (<13 °C), and saltier (∼36 psu) South Atlantic Central Waters (ACAS, Oliveira & Marques, 2007).

Based on the analysis of the diversity and population structure of vibrios, by using a collection of new isolates originated from seawater, sediment, rhodoliths, coral (Scolymia welsii), and previously characterized strains (Chimetto et al., 2009; Alves et al., 2010; Moreira et al., 2013) from different environments located in the SAO, we investigated (i) the habitat spectra of Vibrio populations in the SAO; (ii) whether these vibrio populations were generalist- and/or specialist-adapted; and (iii) if there was connectivity among the benthic-pelagic systems in AB, the largest South Atlantic reef complex. We performed the taxonomic characterization of the isolates using the reliable taxonomic marker gene pyrH (which has higher discriminatory power than 16S rRNA sequences, allowing the distinction of closely related Vibrio species) (Thompson et al., 2005a; Thompson et al., 2005b; Thompson et al., 2007) and inferred ecological associations by using a mathematical model (AdaptML) (David, 2010; Hunt et al., 2008).

Material and Methods

Sources of isolation

In total, 775 vibrio strains, isolated between 2005 and 2010, and identified by means of pyrH sequences, were analyzed. Information on locations and sources are detailed in Table S1 (strain list). The geographic span represents over 3,000 km, including the oceanic SPSPA, AB, and the southeastern Brazilian Saint Sebastian channel (SS) (Fig. 1, map). For the 195 novel isolates, the sources were: sediments from the Buracas (N = 25) (for a full description of the so-called Buracas reef system see Cavalcanti et al., 2013a); rhodoliths (N = 76) (for a full description of the rhodoliths holobiont see Cavalcanti et al., 2013b) from 27 and 43 m depth; seawater from AB, from 10 and 150 m depth (N = 76), including two locations (station 61: closer to shore, and station 65: oceanic), and S. welsii mucus (N = 18), from SPSPA. The remainder 580 strains were obtained previously (Chimetto et al., 2009; Alves et al., 2010; Moreira et al., 2013) and isolated from: corals (Madracis decactis, Mussismilia braziliensis, M. hispida, Phylogorgia dilatata); zoanthids (Palythoa caribaeorum, P. variabilis and Zoanthus solanderi), and fireworm (Hermodice carunculata). These previously characterized data contributed to an increase in the number of habitat categories under comparison as well as their geographic extent, allowing a more comprehensive evaluation of potential ecological and genetic relationships among locations. Sampling permit Sisbio n. 24732-1 was issued by the Ministry of Environment Institute Chico Mendes (ICMBio).

Figure 1 Brazil Map showing the sampling regions.

Microenvironments are highlighted in each sampling site. (A) Saint Peter and Saint Paul Archipelago. Hosts investigated Hermodice carunculata, Scolymia wellsi and Madracis decactis. (B) Abrolhos Bank. Sediment, rhodolith, water, Mussismilia brasiliensis, M. hispida and Phylogorgia dilatata. (C) Saint Sebastian region. M. hispida, Zoanthus solanderi, Palythoa caribaeorum and P. variabilis.

Isolation and preservation of vibrios

Vibrios originating from the water column were obtained from inoculation, performed on board the RV Prof. W. Besnard, in July 2007, in AB. Samples were collected with a rosette system in three depths (10, 75 and 150 m) in station 65 (17°0′36″S; 36°59′56.4″W-oceanic), and at 10 m in station 61 (17°0′3.6″S; 39°0′0″W-on the shelf). Samples from sequential filtration (200 mL) in 3 µm and 0.22 µm filters and aliquots of unfiltered seawater (1 mL) were plated onto the culture medium thiosulfate-citrate-bile salt-sucrose Agar (TCBS) (Oxoid) to obtain vibrios strains. Plates were incubated at 26–28 °C for 48–72 h. Similarly, in independent cruises to SPSPA (00°56′N; 29°22′W) and the Buracas reefs (27 m deep: 17°81′33.0″S; 38°23′74.4″W and 43 m deep: 17°81′39.9″S; 38°24′30.6″W) in 2010, aliquots of coral mucus (S. welsii,) and rhodoliths were surveyed. Rodoliths were crushed and homogenized (0.1 g) in sterile saline buffer (3% NaCl, SSB). Coral mucus was 10-fold serially diluted in SSB. Homogenates (0.1 mL) were plated in triplicates in TCBS at 28 °C. Isolates were purified at Federal University of Rio de Janeiro (UFRJ). The pure cultures were maintained in vials with Tryptic Soy Broth (Oxoid) with 3% NaCl or Marine Broth media, both supplied with 20% (v/v) glycerol, and preserved at −80 °C.

Taxonomic characterization

Characterization of all vibrio isolates was obtained by pyrH sequencing (80F and 530R primers) as described previously (Thompson et al., 2005a; Thompson et al., 2005b). DNA extraction was performed based on Pitcher, Saunders & Owen (1989). PCR sequencing reactions, consensus sequences determination and alignment were performed as in Moreira et al. (2013). Phylogenetic trees were built in MEGA 5 (Tamura et al., 2011). The topology of the tree was based on neighbor-joining method. Distance estimations were obtained according to the Kimura-2-parameter model and the Maximum Composite Likelihood model. Bootstrap percentages were used after 1,000 replications. To analyze the ecological grouping, the multi-FASTA file was converted in MEGA 5 to PHYLIP 3.0 format. The .phy file was used as input to PhyML 2.44 (Guindon & Gascuel, 2003; Guindon et al., 2010) where the suitable tree was generated and again used as input file for the AdaptML software (David, 2010; Hunt et al., 2008). Tree figures were generated using the interactive Tree of Life web application (itol.embl.de) (Letunick & Bork, 2007).

Ecologic grouping of vibrios

Clusters of vibrios’ sequences were obtained with the software AdaptML as described previously (David, 2010; Hunt et al., 2008). In brief, the software combines genetic information embedded in sequence-based phylogenies and information about the ecology, herein the source and place of isolation, in order to identify genetically- and ecologically-distinct bacterial populations. This quantitative model (AdaptML) uses a Hidden Markov Model to predict the phylogenetic bounds of ecologically distinct populations, and their habitat composition (distribution among environmental categories). AdaptML algorithm can account for environmental parameter discretization schemes and is based on the model concept of a habitat (a place and related features that determines microbial distribution). Habitats are characterized by discrete probability distributions describing the likelihood that a strain adapted to a habitat will be sampled from a given ecological state (e.g., at a particular location in the water column or in a specific host). Habitats are not defined a priori but rather learned directly from the sequence phylogeny and ecological data using an Expectation Maximization routine. Once habitats are defined, a maximum likelihood model is used for the evolution of habitat association on the tree (David, 2010; Hunt et al., 2008). The habitat-learning and clustering steps of AdaptML were performed using the default settings. Confident assignments are shown for ecological populations predicted by the model. The model threshold value was set at 0.05 and Photobacterium was used as out-group. The Bootstrap percentages analysis was rerun 100 times with the same phylogenetic tree to verify the stability of the predictions. The circular tree figure was drawn using the online iTOL software (Letunick & Bork, 2007). To prevent numerical instabilities in AdaptML’s maximum likelihood computations, branches with zero length were assigned the minimal observed non-zero branch length: 0.001. Clades supported in 80% of bootstraps were shown. The visualization of the distribution of Vibrio groups in all habitats (Fig. S2) was generated by the online tool Many Eyes (IBM Many Eyes Project; Viégas et al., 2007).

All gene sequences obtained in this study are available through the website TAXVIBRIO (http://www.taxvibrio.lncc.br/). The GenBank accession numbers for the pyrH sequences reported in this study are KC871632–KC871720; KJ154031–KJ154048; EU251514–EU251689; EU716656–EU717075; GU186166–GU186371; KC871598–KC871720.

Results

Taxonomic assignment of the vibrio isolates

The taxonomic characterization of the isolates was mainly based on the phylogenetic position of pyrH gene sequences and its similarities in relation to the closest type strain of Vibrio species. The pyrH gene has shown to be a reliable taxonomic marker for the Vibrio group, even able to discriminate closely related species (Thompson et al., 2005a; Thompson et al., 2005b; Thompson et al., 2007). However, in some cases we also performed a Multilocus sequence Analysis (MLSA) of housekeeping genes and whole genome sequencing (Chimetto et al., 2009; Moreira et al., 2014). Most of the vibrio isolates were V. communis (21.9%), V. mediterranei/V. shiloi (19.7%), V. harveyi (12.4%), V. alginolyticus (9.5%), V. campbellii (7.7%). Other prevailing groups were V. maritimus (4.5%), V. tubiashii (3.5%), V. coralliilyticus (3.1%), V. pelagius (2.5%), V. diabolicus (2.2%) and V. chagasii (1.8%). In addition, 22 strains (2.8%) were identified as candidate new Vibrio species based on 16S rRNA (data not shown) and pyrH gene sequence similarities. Ecological populations predicted by the AdaptML model totalized 19 Vibrio groups clustered accordingly Fig. 2. Most species found in the water column (i.e., V. communis, V. campbellii, V. harveyi, V. maritimus, V. pelagius and V. diabolicus) were also isolated from different invertebrate hosts (Table 1 and Fig. S1). However, some species as V. hepatarius and a unique strain of V. alfacsensis were found only in water samples, whilst V. rotiferianus only in the benthos, coral and zoanthid samples (M. hispida, M. braziliensis, P. dilatata and P. caribaeorum). Some species were retrieved from a single given host. V. sinaloensis was found only in the coral Mussismilia (hispida and M. braziliensis), as well as V. madracius (=V. shiloi-like, Fig. 2) was associated only with M. decactis. V. shiloi was found mainly in association with the fireworm H. carunculata and also with M. hispida in SS. V. furnissii was only associated with H. carunculata. We observed low counts (colony forming units—CFU) in the water column (typically < 102 per mL) compared with the abundance of vibrios in reef waters (up to 104 CFU mL−1) (Bruce et al., 2012) and in the coral mucus (106 CFU mL−1) (Alves et al., 2010; Moreira et al., 2013).

Figure 2 Inferred habitat associations for all ancestors of sequenced Vibrio strains.

The rings surrounding the tree represent the isolation source (outer) and the sampling site (inner) from which strains were isolated. The maximum likelihood assignment of nodes to habitats is shown for all nodes, regardless of the confidence of each prediction. Colored circles on each branch indicate the habitat spectrum assignment (H1-H8) for the node immediately below that branch (see above legend for color scheme). Branch lengths are adjusted to aid visualization and do not represent evolutionary distances. * Highlights the isolation source, water.

Table 1 List of vibrios species found in seawater and at the same time in other hosts investigated in this study.

	M. hispida	P. dilatada	M. decactis	M. brasiliensis	P. caribaeorum	S. wellsi	Rhodolith	P. variabilis	Sediment	
V. communis	X	X	X	X	X	X	X	X	ND	
V. harveyi	X	X	X	X	X	X	X	ND	X	
V. campbellii	X	X	X	X	X	ND	ND	ND	ND	
V. chagasii	X	X	X	X	ND	ND	ND	ND	X	
V. pelagius	X	X	X	ND	ND	ND	ND	ND	ND	
V. diabolicus	X	X	ND	ND	ND	ND	ND	ND	ND	
Notes.

ND, Not detected.

Partitioning of vibrio isolates according to their genetic and ecological similarity

The Vibrio isolates were distributed in 8 habitat spectra. These habitat are a spectrum of environment types over which a given population may be isolated from (Table S2; Fig. 2 and Fig. S1). The three studied areas (i.e., SPSPA, AB and SS) and their distribution in the composition of each habitat spectrum is shown in Fig. S1. Some spectrum of habitat were mainly composed by categories from an unique geographic region as in habitats 1 and 3 (H-1, H-3) (from AB), habitats 2 and 8 (H-2, H-8) (from SPSPA) and habitat 4 (H-4) (from SS). Although habitats 5, 6 and 7 (H-5, H-6, H-7) seemed to be more variable, geographic predominance was observed for H-6, dominated by categories from AB (67%), followed by those from SPSPA (25%) and SS (8%) (Fig. S1). H-1 was mainly composed of strains isolated from water (86%), H-2 from M. decactis (92%), H-3 from M. braziliensis (72%), H-4 from M. hispida (69%), H-5 from Mussismilia (hispida and brasiliensis) (72%), H-6 from rhodolith (44%), H-7 from P. caribaeorum (50%) and H-8 from H. carunculata (94%) (Fig. 3).

Figure 3 Distribution of the environmental categories that compose each of the 8 habitats predicted by AdaptML.

Distributions are normalized by the total number of isolates in each environmental category to reduce the effect of uneven sampling.

Although the observation of these different microenvironments in the SAO was based on a single sampling, the correlation found among the inferred habitat spectra, sampling sites, isolation sources and the associations with all ancestors of the vibrios studied indicated structured populations (Fig. 2). The predominant environmental category in the composition of each predicted habitat spectrum can be clearly visualized in Fig. 3.

In more detail, H-1 was characterized mostly by vibrios from AB seawater (86%) and it was divided into 7 groups occupied by different Vibrio species (Fig. 2 marked with asterisk, and Table S2). V. pelagius (group 1); V. maritimus (groups 2 and 3); V. hepatarius (group 4), and V. communis, V. campbellii and V. diabolicus (groups 5–7). H-2 was composed mostly of isolates from M. decactis–SPSPA (92%) and from AB seawater (5%), V. campbellii (N = 42), candidate Vibrio sp. nov (N = 19) and V. maritimus (N = 15) were the most frequently found species (Fig. S3 and Table S2). Two clusters of V. maritimus, one from AB seawater and the other from SPSPA M. decactis, were clearly distinguished (Fig. S3). H-6 had mainly (57%) vibrios from Buracas—AB (44%: rhodolith, 13%: sediment) and from seawater (6%) (Fig. S3). Species highlighted were V. harveyi, V. communis, V. coralliilyticus, V. tubiashii and candidate Vibrio sp. nov. (Fig. S2). H-7 was mainly represented by the same Vibrio species found in H-6, except for potencial new Vibrio species and the presence of V. rotiferianus. The hosts observed in this habitat  were P. caribaeorum (50%) and M. hispida (12%) both from SS; P. dilatata (23%: Recife de Fora), M. braziliensis (10%: Saint Barbara Island), rhodoliths (3%: Buracas), and water (2%: AB) (Fig. 3 and Table S2).

H-3 encompassed mostly isolates associated with Mussismilia (78%), mainly M. braziliensis (68%) (Roi-Roi reef—AB). The species adapted to this habitat spectrum were V. coralliilyticus, V. harveyi, V. communis, V. sinaloensis and V. tubiashii (Fig. S3). H-4 was mostly represented by benthic animals from SS channel (85%). The main host (72.5%) was M. hispida (69%: SS, 2%: AB). M. braziliensis was also represented (1.5%: AB). Vibrio species observed in H-4 were V. alginolyticus, V. communis, V. campbellii, V. tubiashii, V. chagasii and V. sinaloensis (Fig. S2). H-5 was composed of environmental categories from AB (61%) and SS (37%) (Fig. 3 and Table S2). The dominant category was the host Mussismilia (72%), including both species: M. hispida (46%: AB and SS) and M. braziliensis (26%: AB). V. communis and V. alginolyticus were the dominant species (Fig. S2). H-8 comprised mostly V. shiloi associated with H. carunculata in SPSPA (94%). A few V. shiloi strains were associated with M. hispida in SS (6%) (Fig. 3).

Connectivity among the benthic-pelagic systems in Abrolhos bank

The presence of identical pyrH sequences of vibrios from planktonic and coral reinforce the hypothesis of connectivity (Fig. S3). For instance V. communis (PEL4D from 150 m depth and R-680 from M. hispida, G35, G52 from rhodoliths), (PEL103A from 10 m depth and R-239, R-264 from M. hispida); V. harveyi (PEL36B from 10 m depth and 1DA5 from P. dilatata); V. campbellii (PEL44A from 10 m depth and 42A from M. hispida, PEL45A from 10 m deep and A-391 from M. decactis); V. diabolicus (PEL41D from 150 m depth and 4D2 from P. dilatata); V. pelagius (PEL22B from 10 m depth and 28A2 from M. hispida) and V. chagasii (PEL47A from 75 m depth and PA10 from M. braziliensis and 1DA1 from P. dilatata) (Fig. S3). Samples from water clustered with samples from benthos are highlighted (*) in Fig. 2.

On the other hand, some clusters of planktonic strains seemed to have unique pyrH gene sequences (i.e., V. maritimus group, PEL21 (A, B, C, E and F—station 61, 10 m); PEL102 (A e B), PEL106A, PEL111A, PEL121C, PEL122A, PEL124A and PEL125 (A and B); and V. pelagius group PEL115 (A, B, C, D, E, F, G, H and J—station 65, 150 m) (Fig. S3). The distribution pattern of the main Vibrio species from benthic and pelagic sources in AB based on the evolutionary history inferred by the Neighbor-Joining method with pyrH gene sequences (532 positions) can be visualized in the Fig. S4.

A more targeted AdaptML analysis was performed by dividing the isolation sources in generic two hosts, benthic and pelagic, to explore the extent and significance of the coupling events. It resulted in a very similar cluster distribution of the populations of vibrios. Again, isolates from Abrolhos Bank, both from open water and benthos, were present in the same branches. However, the number of habitat spectra stablished were reduced to six (H1-H6) (Fig. S5).

Discussion

Vibrio population distribution in the Southeastern Atlantic

The predicted habitats spectrum were dispersed along a spatial gradient ca. 3,000 km, from the coastal southeastern SS to the most distant from coast Brazilian archipelago, located above the Equator (SPSPA). Ecologically coherent groups were associated with seawater (H-1), corals [M. decactis (H-2), M. braziliensis (H-3), M. hispida from SS (H-4), both Mussismilia species (M. braziliensis and M. hispida) (H-5)], rodoliths and sediment (H-6), zoanthids—P. dilatata and P. caribaeorum (H-7), and the polichaete H. carunculata (H-8).

The behavior of vibrios showed a wide spectrum, from approaching a true generalist to a strict specialist. The dominant group, the Communis cluster, was present in all habitats spectrum, except for H-8 (mainly composed by the host fireworm), and in 10 out of the 12 samples analyzed. This suggests that these hosts may serve as a reservoir for V. communis’ populations when their abundance in seawater decreases. The exceptions were the zoanthid (Z. solanderi) and the fireworm (H. carunculata). Nevertheless it’s worth to note that only V. alginolyticus inhabited Z. solanderi, according to this study, thus raising the possibility that antagonists among its populations (or the animal itself) could be effective against vibrios. Apart from the exceptions, V. communis populations showed a true generalist behavior. V. communis strains were isolated from multiple independent samples and thus do not represent clonal expansion, suggesting that this may reflect a true habitat switch. Moreover, V. communis appeared to have ecologically diversified, possibly by invading new niches or partitioning resources at increasingly fine scales in a similar way to that of V. splendidus in the northwestern Atlantic coast (Preheim et al., 2011; Hunt et al., 2008). Despite being recognized as a generalist, V. splendidus was not represented in this study, which might reflect its low tolerance to high temperatures (>21 °C) (Materna et al., 2012). The other host that seemed unavailable to V. communis was the fireworm in SPSPA. In contrast to the high diversity that corals harbored, this host was dominated by V. shiloi (n = 143) and some V. furnissii strains (n = 4). V. shiloi was also present in corals (M. hispida) in SS. SPSPA and SS are the extremes of the latitudinal gradient uncovered. Corals and the fireworm appear to define the habitat spectrum for V. shiloi. Both host associations were previously observed in the Eastern Mediterranean (Sussman et al., 2003), suggesting they are stable and that population-habitat linkage is highly predictable for V. shiloi. Moreover, in this survey we found V. shiloi associated with corals, in a human impacted coastal area (SS), and with fireworms in the oceanic SPSPA.

Populations of vibrios in the SAO and those in the northwestern Atlantic showed narrow intersection. In addition to V. splendidus, other vibrios found in the American northern coast were V. rumoiensis, V. alginolyticus, V. fischeri/logei, V. penaeicida, V. superstes, V. aestuarianus, V. ordalii, V. breoganii, V. crassostreae, V. kanaloae, V. tasmaniensis, V. gigantis, and V. cyclitrophicus (Preheim et al., 2011; Hunt et al., 2008). The only common group is V. alginolyticus. It was associated with zooplankton and also displayed free-living style in the coastal northern hemisphere (Hunt et al., 2008). In this study V. alginolyticus was also found in coastal areas of AB and SS, but associated with almost all cnidarians surveyed, and not in the seawater. Variation in host association and lifestyle may reflect genome heterogeneity, possibly due to a large set of flexible genes. In members of the Vibrionaceae, small-scale differences in environmental conditions based on microenvironment and season have been shown to drive lineage adaptation (Hunt et al., 2008) and presumably genome content. A representative study targeting V. alginolyticus genomes (n = 192) from the Chinese coast revealed a high prevalence of mobile genetic elements, including integrating conjugative elements (ICEs), superintegron-like cassettes (SICs), insertion sequences (ISs), and two types of transposase genes (valT1 and valT2). Moreover, BLAST searches and phylogenetic analysis of the ICE, SIC, IS elements and transposase genes showed that the corresponding homologues were bacterial derived from extensive sources, indicating intensive exchange with environmental bacteria (Luo et al., 2012). Indeed, the horizontal gene transfer (HGT) mechanism has played an important role in bacterial evolution, facilitating the origins of bacterial diversity and adaptation to new ecological niches (Wiedenbeck & Cohan, 2011). An important feature shared by the habitats where V. alginolyticus was common is the vicinity to coast, and thus to human activities and nutrient enrichment (SS, AB and Plum Island Estuary, NE Massachusetts).

When we look to the evolutionary history of the generalist and specialist vibrio populations found in this study, two distinct clades defined by MLSA of 8 housekeeping genes can be highlighted: Harveyi and Mediterranei respectively (Sawabe, Kita-Tsukamoto & Thompson, 2007; Sawabe et al., 2013). Harveyi clade is composed by 9 species mainly associate with seawater, salt marsh mud, marine animal and coral mucus. In some cases, distinguishing species and strains within this clade is a hard task for taxonomy, because the presence of recombination among closely related species. Meditteranei clade is composed by 4 species mainly found in habitats as warm seawater and coral mucus (Sawabe, Kita-Tsukamoto & Thompson, 2007; Sawabe et al., 2013; Moreira et al., 2014—V. madracius sp novel). Although, the clades are not phylogenetic very closely related, both possess typically pathogenic species in aquatic environments (Reshef, Ron & Rosenberg, 2008; Ruwandeepika et al., 2010; Ruwandeepika et al., 2011).

We also observed geographic influence in H-4, since it included M. hispida from SS, but not from AB. H-5, 6, 7 and 8 were mainly composed of benthic organisms from combinations of two locations, revealing some connectivity across a spatial scale might also occur. H-5 showed connectivity between AB and SS. H-6 showed connectivity between AB and SPSPA. H-7 showed connectivity between SS and SPSPA.

The influence of temporal dimension in habitat distribution is visualized in Fig. S6. Although not all regions were sampled all years, there was prevalence of 2010 strains in H-2, H-6 and H-8; and of 2007 in H-1 and H-3. On the other hand, strains from 2005, 2006, 2007 and 2010 were present in H-4, H-5 and H-7.

Habitat delineation and taxonomy are congruent

We observed a good congruence between the ecologic grouping generated by the AdaptML and the currently recognized Vibrio species. However, the AdaptML provided further refinement of the species into subspecific groups that may reflect niche partitioning. For instance, we found groups of V. maritimus associated with the seawater and with the coral M. decactis in SPSPA. We defined ecologic groups of vibrios that live in the water column and in association with the benthic organisms. Some species (such as V. harveyi, V. aliynolyticus and V. communis) occupied different habitat spectra, whereas other species (such as V. hepatarius, V. rotiferianus and V. brasiliensis) appeared to be restricted to one habitat (Fig. S2). Some habitat spectra (e.g., H-1) can be defined by multiple species (e.g., V. communis, V. campbellii, V. diabolicus, V. pelagius, V. hepatarius, V. maritimus and V. chagasii). These species are widespread in the water column (up to 150 m depth), representing 86% of the isolates in this habitat spectrum.

The ecological grouping observed in this study suggests dispersal and connectivity among the benthic-pelagic systems in AB. The distribution pattern of the Vibrio species from benthic and pelagic sources in AB based on evolutionary history inferred by using the Neighbor-Joining method of the pyrH gene sequence (Fig. S4) corroborates with this hypothesis. Genetic coherence among the strains from SS, SPSPA and AB also contributes with the coupling idea. Conspecific identical isolates (e.g., PEL4D and R-680, G35, G52; PEL 103A and R-239, R-264; PEL36B and 1DA5; and others), based on pyrH sequences, originated from the pelagic and benthic systems reinforced the idea of connectivity. Even if AdaptML has mistakenly pooled open water specialists and benthic specialists into one population, these identities (of pyrH sequences from benthic and pelagic isolates) are strong evidence of benthopelagic coupling. It is noteworthy identical pyrH sequences from both compartments and among distantly located isolates, since this gene is one of the most divergent among the pool of housekeeping genes employed for vibrios’ MLSA (Thompson et al., 2005a; Thompson et al., 2005b; Thompson et al., 2007). Isolates from both open water and benthic sites were also detected when the AdaptML analysis was based on generic hosts (benthic and pelagic). Moreover, similar cluster distribution of vibrios populations were observed in both parameters analyzed (Fig. 2 and Fig. S5).

In spite of the low CFU counts observed in the water column, we suggest that dispersal through the seawater may be important for the persistence of vibrios in the environment. In reef waters, dispersal may be promoted by the shedding of bacteria by the coral host, as a mechanism to regulate the abundance of associated bacteria (Garren & Azam, 2012). The presence of both strategies in vibrios highlights their adaptation for thriving in both oligotrophic (e.g., water column) and copiotrophic (e.g., coral mucus, organic matter particles) environments, and illustrates the genome plasticity of this ubiquitous group. Furthermore, several vibrios observed in the plankton of the AB may have a pathogenic potential to corals. However, we did not recover some known coral pathogens (e.g., V. coralliilyticus and V. shiloi) in our pelagic survey, suggesting that some vibrio species may have evolved into associated habitats, as the coral holobiont, for example, as observed in the H-8. It was demonstrated that V. shiloi and V. coralliilyticus use chemotaxis to find their coral hosts, by sensing a β-D-galactopyranoside-containing receptor and the metabolite dimethylsulfoniopropionate (DMSP), respectively, both present in the coral mucus (Toren et al., 1998; Garren et al., 2013). V. coralliilyticus employs also chemokinesis and its swimming ability is noteworthy (Winn, Bourne & Mitchell, 2013; Garren et al., 2013). These vibrios may thus have a higher host association frequency. Interestingly, V. madracius retrieved only from the coral M. decactis, which might indicate a new ecological role of this bacterium in this host. This recently described species (Moreira et al., 2014) is closely related to V. mediterranei/shiloi, which is known for pathogenicity.

Influence of benthopelagic coupling in the coral reef health

We observed that several vibrios associated with the seawater and with benthic organisms (corals) formed a cohesive ecologic unity, indicating the connectivity between the benthic-pelagic compartments. Benthic communities obtain their energy through primary production from the benthic compartment and, to a lesser extent, from the overlying water column. Thereupon, the distribution and abundance of planktonic microbes may be dependent on benthic processes, which affect the transfer of organic material between benthic and pelagic systems (Fowler & Knauer, 1986). Bacteria and phytoplankton production are also stimulated by resuspension of nutrients from the seabed into the photic zone, which in turn stimulates zooplankton production, and so on up the food chain (Wainright , 1987). In the present study, we reinforce the power of ecologic theory already developed for the study of vibrioplankton from temperate areas (Materna et al., 2012; Szabo et al., 2013). The genetic connectivity observed among the vibrios originated from the seawater and coral hosts in the SAO illustrates the potential influence of the pelagic system in the coral reef systems health. In a scenario of increasing abundance of vibrios, mediated by higher global oceanic temperatures, the pathogenic potential of some Vibrio groups may lead to increased incidence of diseases in the marine realm. For instance, V. vulnificus implicated in outbreaks were linked to climate change in Israel (Paz et al., 2007), as well as V. parahaemolyticus outbreaks documented in Alaska and linked to the consumption of raw seafood followed by episodes of increased seawater temperature, pinpointing a link between climate change and disease (McLaughlin et al., 2005; Materna et al., 2005; Martinez-Urtaza et al., 2008).

Conclusions

This study was a first attempt to characterize the diversity and the ecological structure of vibrios in several benthic hosts along a latitudinal gradient in the SAO. The occurrence of vibrios from the benthic systems from SPSPA, AB, and SS in the habitats 2, 3 and 4, respectively, reinforces the hypothesis that each benthic system may have its own microbiome. Moreover, populations of V. communis showed a true generalist behavior, whilst V. shiloi was confirmed as specialist, associated to H. carunculata and corals. AdaptML analysis generated a good congruence between ecologic grouping and the currently recognized Vibrio species, with further refinement of the species reflecting niche partitioning. Vibrios might occupy the pelagic and the holobiont habitats, indicating coupling between these microbes and their benthic hosts. The benthic pelagic coupling observed in AB, which is the largest South Atlantic reef complex, may suggest the importance of vibrios in the global ocean health.

Supplemental Information

Figure S1 Distribution of the studied regions in each habitat composition

Distribution of the studied regions [Saint Peter and Saint Paul Archipelago (SPSPA), Abrolhos Bank (AB) and Saint Sebastian channel (SS)] in each habitat composition defined by AdaptML approach. Scale represents percentage.

Click here for additional data file.

Figure S2 Vibrio species diversity in the habitats

Vibrio species diversity in the habitats. The figure shows how the 775 Vibrio strains are distributed in each habitat. The side of the circle represents the proportion of total strains in each group. Others = represents Vibrio species with low abundance found in that group. Figure generated through Many Eyes website (Viégas et al., 2007).

Click here for additional data file.

Figure S3 PyrH tree of vibrios from plankton, rhodoliths and corals

Phylogenetic tree based on the neighbor-joining distance method using pyrH gene sequences showing the relationships among representative Vibrio species from plankton (blue color), rhodoliths (red color) and corals (black). Type strains of Vibrio were included (bold black), Distance estimations were obtained according to the Kimura-2-parameter model. Bootstrap percentages after 1,000 replications are shown. Divergence bar estimated at 2%. Depth is indicated for planktonic strains.

Click here for additional data file.

Figure S4 Evolutionary history inferred by using the Neighbor-Joining method based on 532 positions of pyrH gene sequence in the final dataset. The bootstrap test (1,000 replicates) are shown next to the branches. The evolutionary distances were computed using the Maximum Composite Likelihood method and are in the units of the number of base substitutions per site. The analysis involved 316 nucleotide sequences including only strains from AB region and type strains of each represented species. All ambiguous positions were removed for each sequence pair. White and red circles represent strains from benthic source and blue circles from pelagic.

Click here for additional data file.

Figure S5 Inferred habitat associations for all ancestors of sequenced Vibrio strains from AB region. The rings surrounding the tree represent the isolation source (outer) and the collection point (inner) from which strains were isolated. The maximum likelihood assignment of nodes to habitats is shown for The maximum likelihood assignment of nodes to habitats is shown for clades supported by bootstraps >80%. Colored circles on each branch indicate the habitat assignment (H1-H6). Branch lengths were adjusted to aid visualization and do not represent evolutionary distances.

Click here for additional data file.

Figure S6 Habitat distribution according to sampling time. The values represents percentages.

Click here for additional data file.

Table S1 Strain List

Click here for additional data file.

Table S2 Habitat assignment

Click here for additional data file.

The authors thank Pedro Meirelles comments and technical support of Oswaldo Maia and Milene M.A. Mesquita.

Additional Information and Declarations

Competing Interests

Author Contributions

Field Study Permissions

DNA Deposition

Fabiano Thompson is an Academic Editor for PeerJ.

Luciane A. Chimetto Tonon conceived and designed the experiments, performed the experiments, analyzed the data, wrote the paper, prepared figures and/or tables, reviewed drafts of the paper.

Bruno Sergio de O. Silva and Ana Paula B. Moreira performed the experiments, analyzed the data, wrote the paper, reviewed drafts of the paper.

Cecilia Valle, Nelson Alves Jr, Giselle Cavalcanti and Gizele Garcia performed the experiments.

Rubens M. Lopes analyzed the data, reviewed drafts of the paper.

Ronaldo B. Francini-Filho and Rodrigo L. de Moura contributed reagents/materials/analysis tools.

Cristiane C. Thompson analyzed the data, contributed reagents/materials/analysis tools, reviewed drafts of the paper.

Fabiano L. Thompson conceived and designed the experiments, analyzed the data, contributed reagents/materials/analysis tools, reviewed drafts of the paper.

The following information was supplied relating to field study approvals (i.e., approving body and any reference numbers):

Sampling permit SISBIO n.24732-1 issue by the Ministry of Environment Institute Chico Mendes (ICMBio).

The following information was supplied regarding the deposition of DNA sequences:

All gene sequences obtained in this study are available through the website TAXVIBRIO (http://www.taxvibrio.lncc.br/). The GenBank accession numbers for the sequences reported in this study are pyrH (KC871632–KC871720; KJ154031–KJ154048; EU251514– EU251689; EU716656–EU717075; GU186166–GU186371; KC871598–KC871720).

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
