# Peer review of "Diversity and ecological structure of vibrios in benthic and pelagic habitats along a latitudinal gradient in the Southwest Atlantic Ocean"

_PeerJ, doi:10.7717/peerj.741_

## Round 0.1 · original submission · Major Revisions

-Pay attention to provide enough details about methodology, cfr comment by reviewer 1 about plating and comments by reviewer 2 about the phylogenetic analyses.
-The section on biogeography should be modified. Reviewer 2 has made several pertinent comments about this, that should be addressed.
-You should also address the issue raised by reviewer 2 about including the other parameters (especially temporal variation) in your analysis.

·

Basic reporting

The work is worth reporting on the journal PeerJ. The manuscript is well written in English. Manuscript is consisted of mimimun items.

Experimental design

Experimental design is fine, but species identification policy must be clarified.

Validity of the findings

The manuscript is well construced, the major aims were answered throug the research.

Additional comments

To elucidate how microbial population structured is an ultimate question in microbial ecology, but mostly unsolved yet. Thompson and co-worker have tried to figure out microbial structuring of coral reef microbiota as a model ecosystem. Authors used pyrH gene sequences obtained from 775 vibrio isolates in addition of publically available 580 pyrH gene sequences for population structuring analysis using AdaptML tool. The aims, presence of generalist and specialist with connections between pelagic and benthic niches, is sound, and materials and methods are well designed. However, less description on the species identification policy caused some ambiguous data interpretations on the vibrio structuring in marine ecosystem.

1. L183-184, and so on. It seems unlikely that 1 mL sample was plated completely due to over-inoculation. Authors described the methods in detail.
2. In results. Authors clearly described how isolates were identified on the basis of pyrH gene sequences, especially identification in V. harveyi/V. owensii/V. communis, and V. mediterranei/V. shiloi groups, because fine identification for these species groups needs skillful criteria.
3. Vibrios are currently tried to define an evolutionary unit, Clade, on the basis of multi locus sequence analysis. Total of 22 clades are proposed now. Mapping the data obtained the author’s research on those proposed phylogeny, authors may also interpret how generalist and specialist have had evolutionary history.

Reviewer 2 ·

Basic reporting

The study entitled « Diversity and ecological structure of vibrios in benthic and pelagic habitats along a latitudinal gradient in the Southwest Atlantic Ocean » by Chimetto tonon et al. analyzed the population structure of 775 Vibrio strains. The sampling of these strains was made from 2005 to 2010 at different months and seasons, each years, in water column and different animal hosts. Some of these metadata were combined with a phylogenetic analysis based on pyrH to partition these strains in “habitat spectra” using the AdaptML software. One of the main finding is the coupling between benthic and pelagic Vibrio population in one of their sampling site (Abrolhos Bank).

The introduction is globally interesting and key points are stated, even if some sentences are awkward (see suggestions). The first paragraph (l56-67) calls a lot of concepts but clearly lack citations (there is only one cited work). I suggest more bibliographical references to support the presented ideas.

Moreover, the manuscript would be better by smoothing out some rough sentences. I added in the general comments some part that might beneficiate of some editing.

Experimental design

The main problem in this analysis comes from the sampling. To be clear, strains were sampled during five years, at different seasons, at 8 different sampling sites and in 12 different sources (animal, water and sediment). While there are many variables in this analysis, only two of them that are studied: the sampling site and the isolation source. The temporal dimension of this analysis is not studied here, nor discussed. I think it’s really important to, at least, give some elements justifying that time/season has a minor effect on the presented model. Without this, the conclusions seem limited.

Another thought about the sampling and the analyses. I don’t understand why the authors used all the data to study the benthopelagic coupling. Indeed, sea water was only sampled in AB. I suggest the authors to use only the strains from this sampling site to make the analysis presented in figure S5.

The result part would be easier to understand if the authors treat the benthopelagic coupling in a separate part.

On a more technical point, the phylogenetic inference in material and methods lacks description (method, model used, node support …).

In the legend of the figure 2, the authors wrote “The maximum likelihood assignment of nodes to habitats is shown for all nodes, regardless of the confidence of each prediction” (l764-765). In the study from Hunt et al. cited in this work, the creators of AdaptML used a confidence threshold or 99.99%. The authors should explain why they do not use the same threshold and maybe show in supplementary data the result obtained with this threshold. As this analysis is critical in the manuscript, the absence of statistical confidence is puzzling.

Validity of the findings

The discussion is globally well written and can be understood by non-specialist of Vibrio or marine ecology. However, I don't think it sufficiently adress one the main finding of the study, the biogeographic effect.

Biogeographic trends for vibrios in the Southeastern Atlantic.
Most of this paragraph (l.363-413) is intended to describe the lifestyle/host association related to genetic modifications of Vibrio taxa from this study. While clear and interesting, I don’t see the relation with the biogeographical pattern explained in the title of the section.
The term “geography” and relatives are only written once in the last paragraph of this chapter which is also the only one to discuss the biogeography. This is a problem as it only represents 5 lines out of the 65 lines of this part of the discussion. Moreover, this part only state that there are connectivity between habitat modeled by AdaptML and the link with geographical patterns is unclear and not discussed.

Globally speaking, I think the biogeographical effect is poorly supported by the analysis and discussed. The authors should change the title of this part to reflect his content as biogeography seems to be secondary.

Habitat delineation and taxonomy are congruent.
No specific comments on this part. Some suggestions were added in general comments.

Vibrioplanktonic cells may have an influence in the coral reef health.
I don't think this part of the discussion is of really connected to the scope and objectives of the work.

Globally speaking, I think the biogeographical effect is kind of oversold. The authors should diminish the importance of this finding on every part of the article.

Additional comments

- L82-85: The sentence is too long and difficult to understand.
- L91: “Moreover, increases organic of material used”.
- L93-97: The sentence is too long and difficult to understand.
- L422-423: “We observed a good congruence between the ecologic grouping generated by the AdaptML and the currently recognized Vibrio species” the absence of confidence support in AdaptML analysis (also raised in the Experimental design section) make this assertion elusive.
- L461-462: “and illustrates the huge genome plasticity of this ubiquitous group” the term huge is inappropriate. Genome plasticity is sufficient.
- L472-479: This part mainly discuss the work from Moreira et al., 2014. and is not relevant for the actual work. If the authors want to keep that part I suggest to edit it since the sentence is too long and difficult to understand.
- Supplementary Fig S3: in the first sentence of the legend, delete distance.

---

## Round 0.2 · accepted · Accept

No additional comments - most of the comments raised by reviewers have been addressed.